# Particle retracking algorithm capable of quantifying large, local matrix deformation for traction force microscopy

**Samuel E. Haarman**[1☯], **Sue Y. Kim**[1☯], **Tadamoto Isogai**[2], **Kevin M. Dean**[3], **Sangyoon J. Han**[1,4]*

**1** Department of Biomedical Engineering, Michigan Technological University, Houghton, MI, United States of America, **2** Department of Bioinformatics, University of Texas Southwestern Medical Center, Dallas, TX, United States of America, **3** Department of Cell Biology, University of Texas Southwestern Medical Center, Dallas, TX, United States of America, **4** Department of Mechanical Engineering and Engineering Mechanics, Michigan Technological University, Houghton, MI, United States of America

☯ These authors contributed equally to this work.
* sjhan@mtu.edu

## Abstract

Deformation measurement is a key process in traction force microscopy (TFM). Conventionally, particle image velocimetry (PIV) or correlation-based particle tracking velocimetry (cPTV) have been used for such a purpose. Using simulated bead images, we show that those methods fail to capture large displacement vectors and that it is due to a poor cross-correlation. Here, to redeem the potential large vectors, we propose a two-step deformation tracking algorithm that combines cPTV, which performs better for small displacements than PIV methods, and newly-designed retracking algorithm that exploits statistically confident vectors from the initial cPTV to guide the selection of correlation peak which are not necessarily the global maximum. As a result, the new method, named 'cPTV-Retracking', or cPTVR, was able to track more than 92% of large vectors whereas conventional methods could track 43–77% of those. Correspondingly, traction force reconstructed from cPTVR showed better recovery of large traction than the old methods. cPTVR applied on the experimental bead images has shown a better resolving power of the traction with different-sized cell-matrix adhesions than conventional methods. Altogether, cPTVR method enhances the accuracy of TFM in the case of large deformations present in soft substrates. We share this advance via our TFMPackage software.

## Introduction

Traction force microscopy (TFM) is a soft gel-based assay that reports the spatial distribution of the traction transmitted via cell adhesions of a cell or cells. Taking a pair of images of beads in or on a gel, with and without cell(s), as an input, a TFM software quantifies the gel deformation and reconstructs the traction field using the deformation and the knowledge of a gel's elastic modulus. Albeit persistent improvements in latter, i.e., the traction reconstruction [1, 2], e.g., by better solving the inverse problem of the force-displacement relationship, less attention

**Data Availability Statement:** A GUI-based Matlab software is shared via GitHub at https://github.com/HanLab-BME-MTU/TFMPackage.git or via our

lab homepage, https://hanlab.biomed.mtu.edu/
software.

**Funding:** This work was funded by National
Institutes of Health (https://www.nih.org/)
R15GM135806 (S.J.H.) and F32GM117793 (K.M.
D.). The funders had no role in study design, data
collection and analysis, decision to publish, or
preparation of the manuscript.

**Competing interests:** The authors have declared
that no competing interests exist.

has been paid to the improvement in the gel deformation estimation. The likely reason would
be that the existing methods are adopted from the experimental fluid mechanics [3, 4], which
have been well-established and generated a sufficiently accurate displacement field for a small-
to-intermediate deformation level [2, 5]. However, there has been few studies that have tested
the existing methods for a deformation pattern encoded in the bead images in TFM
experiments.

The deformation pattern represented in the pair of bead images is distinctly different from
ones captured in typical fluid mechanics applications. First, unlike particle image velocimetry
images taken by a camera during fluid motion, the TFM bead images are taken only when cells
are present on the gel or when they are released from the gel, i.e., no intermediate images are
taken during gel relaxation after cell release. Consequently, if cells are on a very soft gel sub-
strate, e.g., < 5 kPa in Young's modulus, or/and cells are decently contractile, the image pair
likely might have recorded a large deformation, e.g., > 4 μm by cellular traction force. More-
over, this deformation is usually highly localized, i.e., concentrated at the locations of cell-
ECM adhesions, thus, the deformation gradient is usually much higher than the flow recorded
by a particle image velocimetry (PIV) experiment. Although not reported in the existing litera-
ture, this large, local deformation is challenging to measure by standard deformation quantifi-
cation methods. To avoid such challenges, most TFM experiments have used 'stiff-enough'
substrates to be able to track the deformation without a problem [6–8]. This limited usage has
either reduced the range of stiffness for cells' mechanosensitivity studies or missed potentially
large force from a prominent integrin-based adhesion even in the moderate stiffness environ-
ment. However, systematic investigation on why the existing methods fail to detect the large,
local deformation and how those methods can be improved algorithmically have not been
investigated yet.

The first and most frequently-used deformation quantification technique is PIV [3].
PIV relies on the image cross-correlation to estimate the likely displacement of a template
image, from the bead image taken in unstressed gel configuration, on the bead image of
the 'stressed' gel, based on the grid points of the reference bead image [9]. Later, PIV was
further evolved to have more precision and resolution with adopting iterative image defor-
mation scheme [10, 11]. However, a few downsides and limitations of PIV have also been
identified and discussed [9, 12]. First, PIV has an inherent averaging effect, i.e., the dis-
placement vector of one image patch is an outcome of an average displacement of all parti-
cles within the finite template size [11, 13]. Moreover, even when a smaller template is
used to contain only a few beads in it, the resultant displacement per template is again
biased by usually off-centered bead signal in the template area [12]. This inherent limita-
tion of PIV can be a huge problem for TFM because even small error in the displacement
field can be greatly amplified during the solution to the inverse problem, which is usually
ill-conditioned [2].

To work around this, the correlation-based particle tracking velocimetry (cPTV) has been
developed [14, 15] and further improved for TFM [2]. The cPTV uses the image cross-correla-
tion to find a displacement but it does it based on the individual bead locations identified by
fitting Gaussian-mixture model to a finite-sized bead image window [16]. Owing to a less bias
associated with the off-centeredness, cPTV is deemed to be a better-suited tracking method
than the conventional PIV for TFM purpose [5, 14]. However, as cPTV also uses the image
cross-correlation, it still fails to capture a large, local deformation due to poor correlation in
such displacements.

In this paper, we investigate performance of existing PIV and cPTV methods on simulated
bead images exhibiting a known displacement field out of a force field showing a large, local
traction (~10 kPa). We identify that the image correlation score from a bead to a true, large

displacement is smaller than a score to other candidate positions, leading to failure of displacement tracking by all existing PIVs and a cPTV. To circumvent this failure, we develop an algorithm, named cPTV-Retracking, that seeks to choose a deformation vector based on its similarity to neighboring vectors in terms of magnitude and direction, even when it is a non-maximal cross-correlation peak. We show that the new algorithm is able to track 92% of the large vectors that have been missed or poorly estimated by the conventional methods, resulting in better overall accuracy. Analyzing experimental images of beads coated on a silicone gel with 4 kPa using the new method, we show that a mouse embryonic fibroblast exerts 40% more traction force via long-enough focal adhesions than forces analyzed by conventional methods. Correspondingly, the new method increases the resolving power of traction force per maturation status of the cell-ECM adhesions. Taken together, our novel displacement tracking enables to capture large, local displacement vectors, which overall increases the resolution and accuracy of the TFM traction distribution.

## Methods

### Simulation of large displacement field and synthetic bead images

Traction distribution where there is a large force, e.g., 12 kPa in magnitude, with several other small force spots, is created in a 512x512 pixel field (Fig 1A), from which the displacement filed is calculated via integrating Boussinesq solution-based Greens function, with 8 kPa of Young's modulus and 0.5 of Poisson's ratio (Fig 1C) [2]. For testing multiple tracking algorithms, a synthetic bead image was created by randomly super-imposing 8000 2D Gaussian functions in 512x512 pixel area (Fig 1B), which was used as a reference bead image of undeformed configuration. Then the calculated displacement field was applied to individual 'beads' to create a bead image of deformed configuration (Fig 1B).

### Particle image velocimetry (PIV) methods

PIV builds a velocity field by dissecting an area of interest into the same-sized interrogation areas and determining velocity vector per interrogation area using image cross-correlation, extensively used in fluid mechanics and aerodynamics [12]. The basic interrogation method has evolved further with using normalized cross-correlation, windowing, peak deformation, iterative image deformation, Gaussian low-pass, interrogation overlapping, and vector interpolation [17]. Accordingly, multiple user-friendly PIV methods were implemented and shared. First, PIVSuite, a Matlab command-line-based PIV package [18], was adopted and used for our study by incorporating it into our TFM package [2]. PIVSuite uses multi-pass, multigrid algorithm, with image deformation, inspired by another PIV software, PIVlab [19]. Second, mpiv, another Matlab-based PIV toolbox [20], was used as the second PIV method for the study. It adopts Minimum Quadratic Difference (MQD) algorithm, recursive super-resolution PIV, and subpixel analysis as well as image transformation. Finally, Tseng's PIV, an ImageJ-based plugin used ultimately for TFM [21], was used as the third PIV software to be tested. Based on another well-established PIV software, JPIV [22], it adopts an iterative scheme where interrogation is done via normalized cross-correlation per iteration and incorrect displacement vectors are filtered out based on correlation score. The three PIV methods were chosen for comparison because they were easily accessible and sufficiently advanced. To be consistent, all PIV methods used the following parameters (Table 1). Some of these parameters, e.g., template window length, window step size, and a type of filter, were chosen after performing a sensitivity test using bead images containing an intermediate level of displacement (S1 Fig). In short, the sensitivity tests showed that small-enough final template window length (e.g., 16 pixel), step size (8 pixel) and Welch filter produce the best tracking results. Accordingly, we

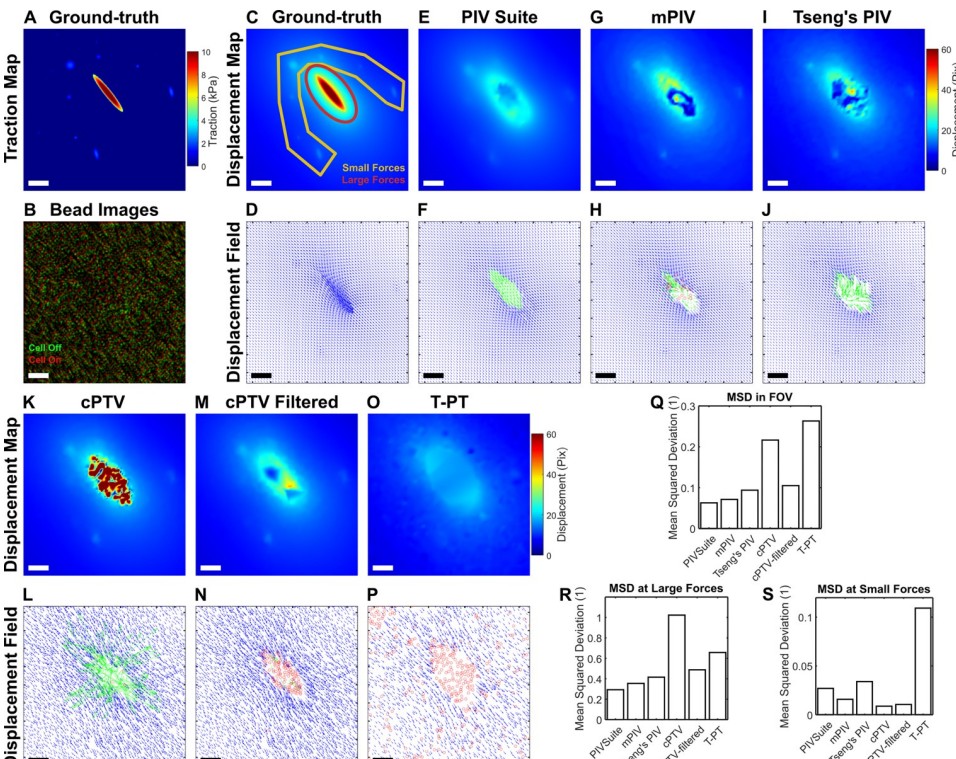

**Fig 1. Typical PIV methods, a topology-based tracking and the current cPTV fail to track a large, local deformation.** (A) Ground-truth traction map, designed with a large, local traction force in the center, mimicking the force from a large focal adhesion. (B) A color-merged image of a synthetic bead image (8000 beads on 512x512 pixel area) in a relaxed configuration (red) and another image of beads where displacements in A is applied (green). (C,D) Ground-truth displacement magnitude map (C) and vector field (D) calculated from the ground-truth traction distribution. Yellow polygon overlaid represents the region of displacement by the small forces whereas the red ellipse represents the region of displacements by the large force. (E-P) Displacement magnitude map (E,G,I,K,M,O) and corresponding vector field (F,H,J,L,N,P) of the pair of bead images from panel B, tracked by multiple PIV methods such as PIV Suite (E,F), Tseng's PIV (G,H) and mpiv (I,J), a correlation-based PTV, or cPTV (K,L), cPTV where vector outliers are filtered (M,N), and a topology-based particle tracking, or T-PT method (O,P). Red circles in the vector fileds represent the seed points that have failed to track the deformation. Green arrows represent falsely-tracked vectors that show more than 10% mean-squared-deviation. Scale bars in A-P: 5 μm assuming 108 nm/pixel. (Q-R) Bar plots of the mean-squared-deviation (MSD) for the measured displacement fields by the six tracking methods, quantified over the entire field of view (Q), large force area (R), and small force area (S).

assumed that other PIV methods (mpiv and Tseng's PIV) perform similarly as PIV Suite with these chosen parameters. The cross-correlation was performed in the frequency domain using Fourier transform.

These three PIV software were used for comparison with the cPTV for TFM purpose, assuming that these were advanced enough as other undiscussed, but well-established, PIV software such as OpenPIV [23], PIVlab [19], MatPIV [24], OSIV [25] and PPIV [26].

**Table 1. Parameters used for PIV methods.**

| Method | Multipass iterative? | No. of Passes | Window lengths | Maximum Displacement | Window Step Size | Filter | Interpolation |
|---|---|---|---|---|---|---|---|
| PIVSuite | Yes | 4 | 64 64 32 16 | 60 | 8 | Welch | Yes |
| mPIV | Yes | 3 | 64 32 16 | 60 | 8 | None | Yes |
| Tseng's PIV | Yes | 3 | 64 32 16 | 60 | 8 | None | No |

## Topology-based particle tracking (T-PT)

To further improve the robustness of the investigation, a reputably published particle tracking method was included alongside the PIV methods mentioned above. The T-PT algorithm utilizes the spatial relationship between detected particles and their neighbors to establish feature descriptions that allow the particles to be tracked across frames [27]. This is in contrast to the image correlation-based methods used in PIV and, as such, the parameters required for T-PT are unique. The window size selected for T-PT was 13x13, 16 initial neighbors and 8 retained neighbors were used with three concentric spheres to generate the feature descriptors. An outlier threshold of five was used for the universal median test.

## cPTV algorithm

Traditional PTV identifies centers of individual particles from a pair of images and match the locations per frame by a suitable pairing algorithm [28]. It has been found to make tracking more reliable and robust when a cross-correlation is used to provide information as a predictor-corrector [12]. Accordingly, cPTV, which combines both PIV and PTV, has been used in the field of TFM [2, 14]. cPTV identifies individual beads using Gaussian-mixture model [29], then apply the normalized cross-correlation,

$$S(u_x, u_y) = \frac{\sum_{i,j=-L}^{L}(I_{ref}(i,j) - \overline{I_{ref}})(I_{def}(i + u_x, j + u_y) - \overline{I_{def}})}{\sqrt{\sum_{i,j=-L}^{L}\left(I_{ref}(i,j) - \overline{I_{ref}}\right)^2}\sqrt{\sum_{i,j=-L}^{L}\left(I_{def}(i + u_x, j + u_y) - \overline{I_{def}}\right)^2}}, \qquad (1)$$

where S is the cross-correlation score, $u_x$ and $u_y$ are the interrogation shift in x- and y-directions, $I_{ref}$ is the reference bead image, $I_{def}$ is the bead image taken at deformed configuration (by a cell), L is the half-length of a template window, $\overline{I_{ref}}$ and $\overline{I_{def}}$ are the average pixel intensity of $I_{ref}$ and $I_{def}$, respectively. The interrogation shifts, $u_x$ and $u_y$, travel up to a certain maximum distance, $u_{x,max}$ and $u_{y,max}$, both in positive and negative directions per axis. Usually a single maximum distance, $u_{max}$, is used for both directions, *i.e.*, $u_{x,max} = u_{max}$ and $u_{y,max} = u_{max}$. Less than 30 pixels in width and height was used to find the most likely displacement. To find if the correlation score maximum is a true maximum, a heuristic significance criteria, where the global maximum is compared with the secondary local maximum with a certain ratio, was used as previously done [2, 15]. Briefly, the significance criterion computes the ratio of the amplitude of each peak, e.g., secondary, tertiary peak, etc., with respect to the global peak. With higher significance criterion ratio (e.g., 0.9), the criterion becomes more generous in verification of the maximum score and associated displacement. A small significance criterion ratio (e.g., 0.3) makes the test stricter, picking up a set of only strongly obvious displacement vectors. After finding the best candidate displacement in a pixel level, a subpixel displacement is estimated, by default, using a parabolic surface fitting. When a bead location does not pass the significance criteria, the position is marked missing by assigning 'NaN' values. More detailed description is available in our previous publications [2, 30]. Outliers of displacement vectors were filtered out using vector-median-based filtering where each vector is compared with median vector of a neighborhood of vectors around it [9].

## cPTV-retracking (cPTVR) algorithm

Overall flowchart about cPTV-Retracking is shown in Fig 3. The algorithm assumes that the first-time cPTV is performed with a high significance criterion and a strict filtering criterion, from which 'seed points' consisting of failed tracking and filtered-out locations, are determined. Looping through these seed points, neighboring vectors with a certain search radius

are gathered per location, from which statistics such as mean and standard deviation is calculated. Then cross-correlation score is calculated over a range of displacement magnitude and angle from their medians. All the local maxima within the range are obtained from which the global maximum is first compared with the second maximum local maximum using the significance criteria. If the global maximum passes the criteria, it is confirmed as the displacement. If not, a local maximum, not necessarily global or second maxima, closest to the neighbor model vector is obtained. A criterion that checks the closeness is developed as when the distance is less than two standard deviations of the neighboring vectors, and the angle is less than a standard deviation of orientations of the neighboring vectors. This criterion is adjustable by user input after checking the retracked output. If the peak passes this criterium, the candidate is determined as the displacement for the position. If not, a new correlation score with a smaller interrogation window is calculated, from which global or local maxima are assessed as done for the initial assessment, iteratively. Further details are described in the results section.

## Mean squared deviation

The mean squared deviation (MSD) was used to determine how much the tracked displacements are deviated from the ground-truth displacement field and was calculated as,

$$MSD = \frac{\sum_{i=1}^{N} |\overrightarrow{V_{g,i}} - \overrightarrow{V_{m,i}}|}{\sum_{i=1}^{N} |\overrightarrow{V_{g,i}}|}, \tag{2}$$

where N is the number of the measured displacement vectors, $\overrightarrow{V_{m,i}}$ is the i-th vector of the measured displacement field, and $\overrightarrow{V_{g,i}}$ is the i-th vector of the ground-truth field from the same, matching origin as $\overrightarrow{V_{m,i}}$. The vectors in the region of interest, e.g., the entire field of view, the large force area, and the area of small forces, are separately entered into MSD calculation to assess tracking accuracy of each tracking method.

## Accuracy

MSD increases as the measured displacement becomes more inaccurate. To create an accuracy-measuring metric that increases with increasing accuracy, we defined a metric called 'accuracy', *Acc*, which is calculated as:

$$Acc = \frac{1}{N} \sum_{i=1}^{N} \left( 1 - \frac{|\overrightarrow{V_{g,i}} - \overrightarrow{V_{m,i}}|}{mean(|\overrightarrow{V_g}|)} \right), \tag{3}$$

where each variable is explained in the section for MSD. Accuracy of 1 means that all ground-truth vectors are correctly tracked whereas Accuracy of less than 1 means at least some vectors are inaccurately tracked or missed.

## Traction reconstruction

The obtained displacement fields were transformed into the force fields by solving the inverse problem of force-displacement relationship, or a Greens function. Boussinesq solution was used as a proper Greens function [31], and fast boundary element method (FastBEM) with L2-norm-based regularization was employed for the force reconstruction, with regularization parameter chosen by L-curve, L-corner method [2].

## Cell culture

Human Osteosarcoma U2OS cells were cultured in Dulbecco's Modified Eagle's Medium supplemented with 10% fetal bovine serum (Sigma; F0926-500ML) and maintained in an incubator at 37˚C and 5% $CO_2$. All cells were tested for mycoplasma. Cells were labeled with mNeonGreen-paxillin via lentiviral infection.

## Total internal reflection fluorescence microscopy (TIRF) for TFM and adhesion imaging

Cells were plated on 4kPa high refractive-index silicone gel coated with 40 nm-diameter fluorescent, AF647-coated, beads. They were imaged under a DeltaVision OMX SR (General Electric) ring-TIRF with a 60x oil-immersion objective and two sCMOS cameras at 37˚C, 5% carbon dioxide, and 70% humidity. Minimum laser power was used (usually between 0.2–2%) to prevent photo-toxicity. One camera was used to take bead images with Cy5 laser, and the other camera was used to take paxillin images with 488 nm laser. A reference bead image was additionally taken after releasing cells with high-dose (0.25%) trypsin. Bead images were entered into TFM package to be processed for displacement tracking and traction reconstruction.

## Focal adhesion segmentation and nascent adhesion detection

Matured focal adhesions (FAs) were segmented as previously described [2]. Briefly, adhesions from images with paxillin staining were segmented using a combination of Otsu and Rosin thresholds. Segmented areas larger than 0.2 μm$^2$ were considered focal contacts (FCs) or FAs, based on the criteria described by Gardel et al [32]. Diffraction-limited nascent adhesions (NAs) were detected using the point source detection used in single particle tracking [29], which consists of filtering images with a Laplacian of Gaussian filter, detecting local maxima and fitting them with an isotropic Gaussian function (standard deviation: 2.1 pixel). Outliers were removed using a goodness-of-fit test ($p = 0.05$).

## Results

To test if existing PIV methods are able to track a large, local deformation, we created a ground-truth traction field consisting of a large traction, i.e., >10 kPa, and a few spots of small tractions (Fig 1A). We built a synthetic bead image by randomly super-positioning 2D Gaussians (Fig 1B, red) and applied the ground-truth displacement field, calculated by using the Boussinesq solution assuming E = 8 kPa (Fig 1C and 1D, see Methods), to the individual beads to create another bead image, mimicking cell adhesion presence (Fig 1B, green). We processed the image pair using the three different, widely-used, available PIV software, namely PIVSuite, Tseng's PIV, and mpiv, to track the displacement field (see Methods, Fig 1E–1J). For valid comparison, we used the same template width and grid spacing for all three PIV methods, which were chosen via a sensitivity test of PIV Suite with moderate deformation (S1 Fig, See Methods). The three methods resulted in slightly different displacement fields overall, in terms of magnitude, direction and spacing between adjacent vectors. However, one common trend is that they failed to track the large displacement vectors around the center region (Fig 1F, 1H and 1J). Specifically, PIVSuite produced systematic underestimated displacements in the center of large force regime (Fig 1F, green vectors) while Tseng's PIV resulted in missed several seed locations for displacement tracking (Fig 1H, red circles). In a similar manner, mpiv missed more locations in large force region for displacement estimation (Fig 1J). We also tested a performance of cPTV by applying it to the same pair of the synthetic bead images.

cPTV identifies individual beads, from which an interrogation is performed by the cross-correlation. A significance criterion is used to determine whether the global maximum of the correlation score is significantly higher than the second largest local peak (see Methods). When a strict criterion is used, many vectors in the large force regime were missing as in mpiv. When a more generous significance criterion is used, many large vectors in the large force region were obtained (Fig 1K), but they were incorrect (Fig 1L). However, it was noticeable that cPTV was able to distinctly capture the displacement vectors in the region of low forces (Fig 1K). Filtering out vector outliers removed most of large, wrong vectors in large force region but preserved well-tracked small vectors, e.g., on the small force area (Fig 1M and 1N). To test if the large, local deformation is trackable by a tracking method that does not rely on image-correlation, we applied a recently developed, topology-based particle tracking (T-PT) method [27]. Unexpectedly, T-PT has resulted in more underestimated displacement field where large vectors were also completely untracked (Fig 1O and 1P).

To assess the tracking accuracy, we used the mean-squared-deviation (MSD) over the entire field of view of the displacement field resulted from each tracking method against the ground-truth displacement field (Fig 1Q). T-PT showed the most amount of deviation followed by cPTV, filtered cPTV, Tseng's PIV, mpiv and PIVSuite. A large portion of deviation came from the missing or wrong vectors at the large force area (Fig 1F, 1H, 1J, 1L, 1N and 1P). Quantifying MSD at only the large force area (Fig 1C, red ellipse) indicated that cPTV created the largest deviation, which was reduced by the outlier filtering (Fig 1R). The low MSDs by mpiv, Tseng's PIV and PIVSuite at the large force area were likely attributed to the interpolation of vectors in their algorithms (Fig 1F and 1H–1J). At the small force area, however, cPTV, with or without filtering, has shown the exceptionally lowest MSD against all other PIV variants and T-PT (S1 Fig). This result demonstrates the superiority of cPTV to PIV methods and topology-based tracking, at least for small-force-driven deformation, as the interrogation starts at the center locations of individual beads whereas PIVs do from the centers of the gridded windows. Together, these data demonstrate that all current PIV, T-PT, and cPTV methods fail to track a large, local displacement field, and if cPTV gains further accuracy in the large displacement tracking, it will be the method of choice for force-induced displacement quantification.

To investigate the reason behind the failure in large deformation tracking, we inspected the correlation score and its global vs. local maxima for small and large deformation cases (Fig 2). For the small displacement, for example, we have picked a vector position whose true displacement is less than 30 pixels and plotted the cross-correlation score map (Fig 2A), equivalent to a top view of the surface plot of the score. Then we depicted the shift position of the global maximum, i.e., the measured displacement (Fig 2A, a red circle), as well as the shift position of the ground truth vector (Fig 2A, a purple circle). The two circles being at the same location for the small displacement represents that the image interrogation finds the vector successfully. To check the image matching visually, we also directly mapped the center location of the bead of interest on the reference image (Fig 2B) and its shifted position on the image in the deformed configuration (Fig 2C). Cropping and enlarging the template windows in the reference (undeformed) image and at ground-truth and measured positions in the bead image of the deformed configuration confirmed that the template images in both ground-truth and measured positions are identical to that in the reference image, and so was the correlation score (Fig 2D).

Applying the same visualization pipeline to a large displacement vector showed that ground-truth and measured vectors are at different positions on the correlation score map (Fig 2E), indicating that the ground truth is not at the global maximum position for the large, local deformation. Mapping and zooming the template images in the reference and deformed configurations showed that the ground-truth displacement has much poorer correlation with

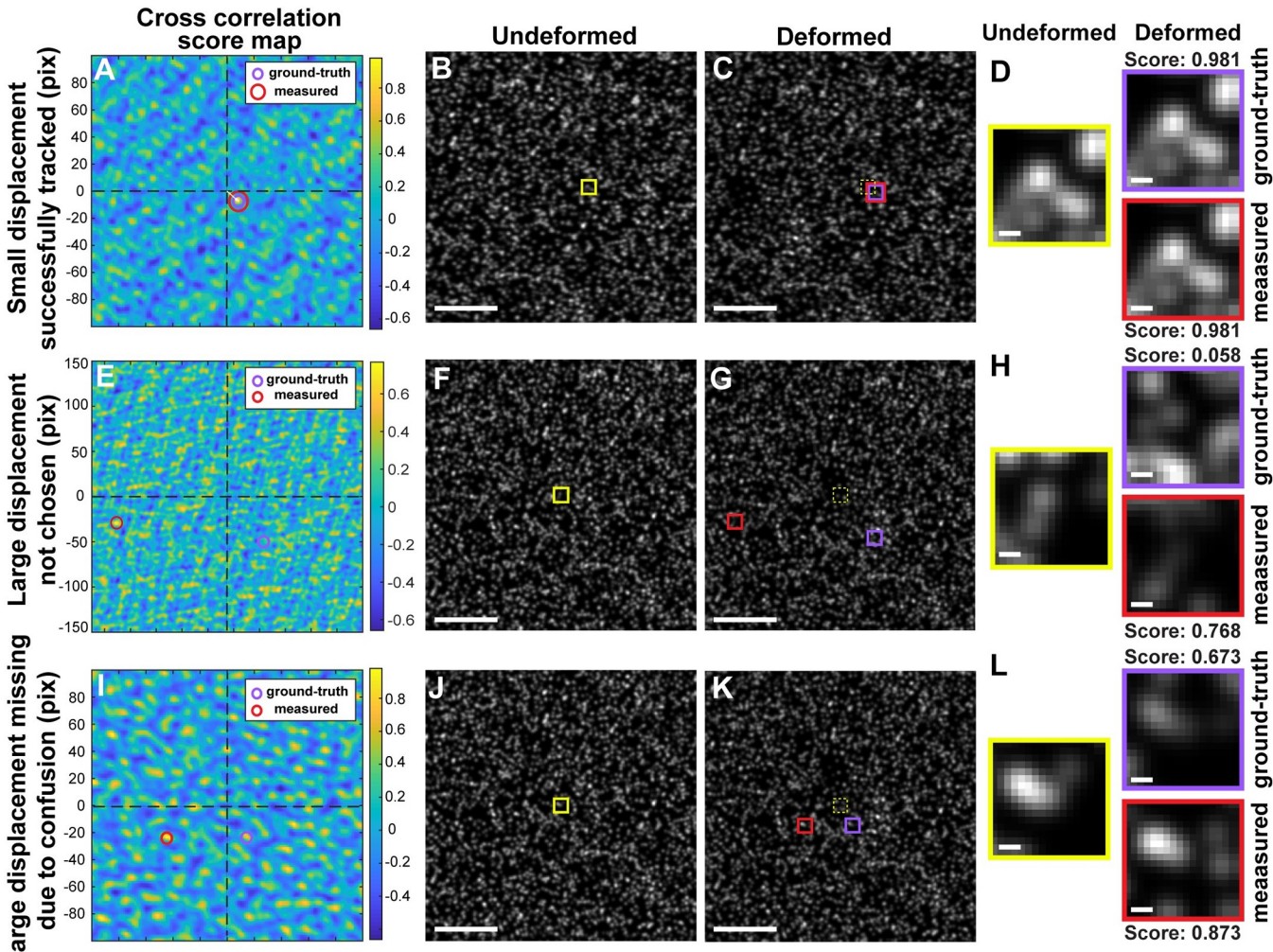

**Fig 2. Large displacement shows a correlation score smaller than global correlation maximum.** (A-L) Cross correlation score maps (A, E, I), the synthetic reference images of the reference (undeformed) configuration with a yellow box indicating the template window around the center position of a bead of interest (B,F,J), the bead images of the deformed configuration with a red box indicating a shifted location of the measured displacement, a purple box indicating a shifted location of the ground-truth displacement, and a dotted white box indicating the original unshifted position of the template window (C,G,K), and the cropped and enlarged views of the templates with boxes whose color matches those in the bead images (D,H,L) for a small (<10 px in magnitude) displacement representative vector (A-D), a large (>50 px in magnitude) displacement vector (E-H) and another large (~30 px in magnitude) displacement vector that was determined to be missing (I-L).

the reference template than with the global maximum by more than ten folds (Fig 2F–2H). Similarly, for a vector that was determined as missing due to failure to pass the significance criteria, we identified that the correlation at the ground-truth displacement was not the global maximum (Fig 2I). The correlation score of the ground-truth displacement was not as low as in the larger displacement case in Fig 2E–2H, as the cropped image also shows also similarity with the reference template image (Fig 2J–2L). However, the fact that the score is smaller than the global maximum but still large enough made the vector fail to pass the significance criterion. Together, this investigation shows that the large displacement caused by a large local traction is unable to be tracked via cross-correlation due to the poor correlation specific to TFM application.

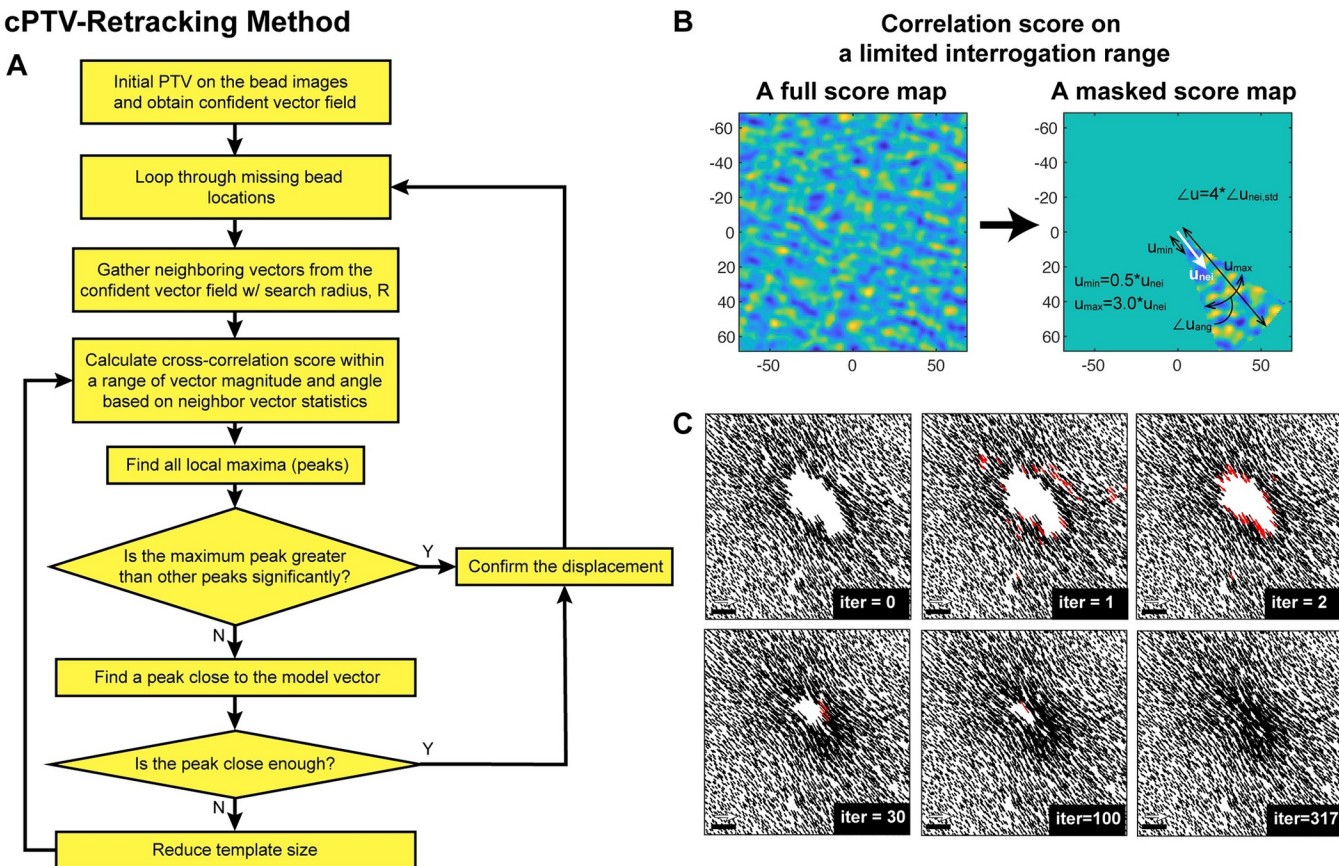

**Fig 3. cPTV-Retracking algorithm.** (A) Flowchart of the main algorithm of cPTV-based retracking. See the main text or Methods section for detailed description of this chart. (B) An example of a masked correlation score map (right) compared to a full score map (left), which is used in step 4 of the flowchart. A white arrow on the masked score map, $u_{nei}$, is the averaged neighboring vector. The range of the radius of the mask was determined as $0.5^*|u_{nei}| \leq |u| \leq 3^*|u_{nei}|$, and the range of the angle was deteremined as $mean(u_{nei})-2^*std(u_{nei}) \leq u \leq mean(u_{nei})+2^*std(u_{nei})$ where the symbol, , represents the orientation of each vector. (C) A series of displacement vector fields at representative iterations, showing how progressively missing vectors are found. Red vectors represent the newly-found vectors at the specific iteration. At zeroth iteration, the field is filtered by a vector median filter with a strict threshold so that the retracking is performed based on the confident neighboring vectors. The last field at iteration = 317 is the result from the last iteration, i.e., when there was no single retracked vector for 30 consecutive iterations.

From this simulation experiment as well as our TFM force reconstruction experience from the experimental bead images, we noticed that failures in displacement tracking occurs exclusively in the large vectors due to concentrated forces. We also learned that the failed large displacement vector has at least the similar direction as the neighboring vectors, despite the poor cross-correlation score. We have thus implemented a new algorithm, referred to as cPTV-Retracking, that tracks the missed or filtered-out vector positions again using information from the well-tracked neighboring vectors (Fig 3). The algorithm is based on the output displacement field from the initial cPTV for the entire field of view in the bead image pair, with a strict significance criterion to produce only well-tracked vectors, from which we identify bead locations that has missed tracking. Per missing location, the algorithm collects neighboring vectors with a certain search radius, from which the average and standard deviation of the magnitudes and the orientations are calculated. Then the cross-correlation score is calculated over the limited range in the magnitude and the angle based on the average neighboring vector. To accommodate the tendency that large vectors are usually missing and need to be retracked, a more generous upper limit is defined compared to a lower limit (Fig 3B). From the new score map,

local maxima are found and compared with the global maximum using the same significance criteria. We found that the reduced number of local maxima increases a chance of the global maximum passing the significance criteria. If it does not pass the criterion, all the local maxima are compared with the 'model vector', which is built from the median neighboring vector instead of the mean. The median was chosen because we observed the neighboring vector magnitudes followed non-Gaussian distribution where there is a consistent bias toward 'larger' magnitude and we expect the missing vectors are at least equal or larger than the neighbors. To test whether a candidate displacement is close enough to the model vector, i.e., thus selectable as the displacement vector for the location, we used one standard deviation of both magnitude and angle within which we choose the vector. If the candidate vector does not pass this test, the algorithm reduces the template window size gradually, e.g., by 2 pixel in width and height, and iterate the candidate vector identification process (Fig 3A), until the template window reduces it size into a size that can contain almost a single bead. Upon successful finding, the found vectors become new neighbor vectors for retracking of another missing bead position. Fig 3C shows an example of gradual 'filling-in' of retracked vectors from the existing confident vectors at several selected iterations.

To make sure if the median-based model vector of the local neighboring vectors is truly representative as a model vector, we introduced a new variable, i.e., the enlargement factor, to adaptively increase the magnitude of the model vector above the median. Fig 4A summarizes the related algorithm to use the enlargement factor for finding larger displacements. It begins with initializing the factor as one and obtain the optimal search radius to con contain at least 3 neighboring vectors. Then we loop through positions that have sufficiently close-enough neighbors for retracking as described in Fig 3A. At the initial retracking, we use just median of the neighboring vectors as a model vector. When there are no more successfully tracked vectors, then the algorithm increases the minimum number of neighboring vectors for the search radius determination and performs the retracking. If it does not generate any retracked vectors, the algorithm decreases the template window length by 2 pixel and increases the enlargement factor by 10% and retrack the missing locations. Applying the enlargement factor increased the chance of finding likely large vectors compared to retracking with only the median vector (Fig 4B and 4C), as expected. To make sure if the newly-found large displacements help improve the tracking accuracy, we measured MSD of the displacement field and compared with the field from cPTVR with the median model vector and the field cPTVR with filtering (Fig 4D). We found that the MSD value of cPTVR with the enlargement factor, or cPTVR-EF, is much smaller than the other two results in both entire field of view (Fig 4D) or specifically in the large force area (Fig 4E). Comparison of the accuracy (see Methodss) between PIV methods and the cPTVR variants confirmed that cPTVR with both median and the enlargement factor exhibited much higher, e.g., >90%, accuracy than other PIV methods, where cPTVR with the enlargement factor showed the highest (~92%) accuracy. Together, these data suggest that a model vector larger than a median increases the rate of finding large displacements in the TFM setting.

To seek whether the retracked displacement field actually leads to a more accurate traction field, we reconstructed the traction field out of the displacement field tracked by cPTVR and compared it with the ground-truth traction field along with tractions produced based on displacement fields by other existing PIV and earlier cPTV methods (Fig 5, *See* Methods). Compared to the ground-truth (Fig 5A and 5B), we found that the traction fields by all PIV methods, i.e., PIV Suite, Tseng's PIV and mpiv, and T-PT method, were largely underestimated or not colocalizing with designed force locations (Fig 5C–5J), most likely due to their failure in large displacement tracking. cPTV without retracking also resulted in the traction field mostly missing large forces whereas it detected small forces well (Fig 5K and 5L).

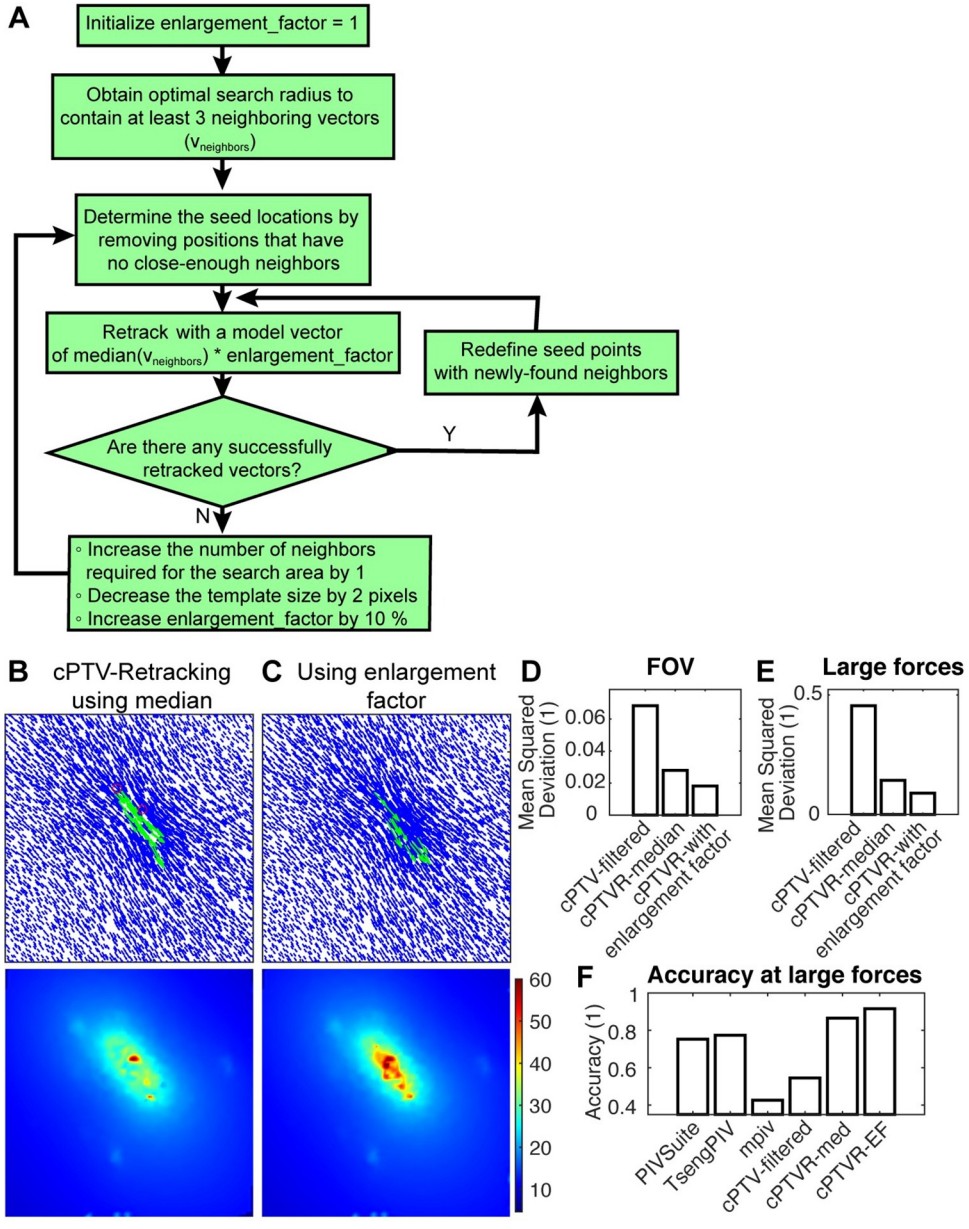

**Fig 4. Containing the enlargement factor increases the possibility of identification of the large displacement vectors and the accuracy of the tracking.** (A) A flowchart of an algorithm that adaptively retrack displacement vectors using the enlargement factor. See the main text for the details. (B,C) Final displacement fields (top) and associated displacement maps (bottom) resulted from cPTVR with a median vector as a model vector (B) and with a progressive enlargement in the model vector (C). Green arrows represent vector outliers determined by ones showing more than 10% MSD. (B) was converged in 317 iterations whereas (C) converged in 119 iterations. (D,E) Bar plots of MSD for the measured displacement fields in the entire field of view (D) or in the large force area defined in Fig 1C (E). Note that cPTVR with enlargement factor has much smaller MSD than the filtered cPTV or cPTVR with a median model vector. (F) A bar plot of the accuracy at the large force area between three PIV methods and three cPTV methods.

cPTV-Retracking, when performed with a median as a model vector, showed small improvement in rescuing large forces, but it did so insufficiently by missing many larger force vectors (Fig 5M and 5N). However, cPTVR performed with the enlargement factor resulted in the traction field in which much more large force vectors were reconstructed as well as small force

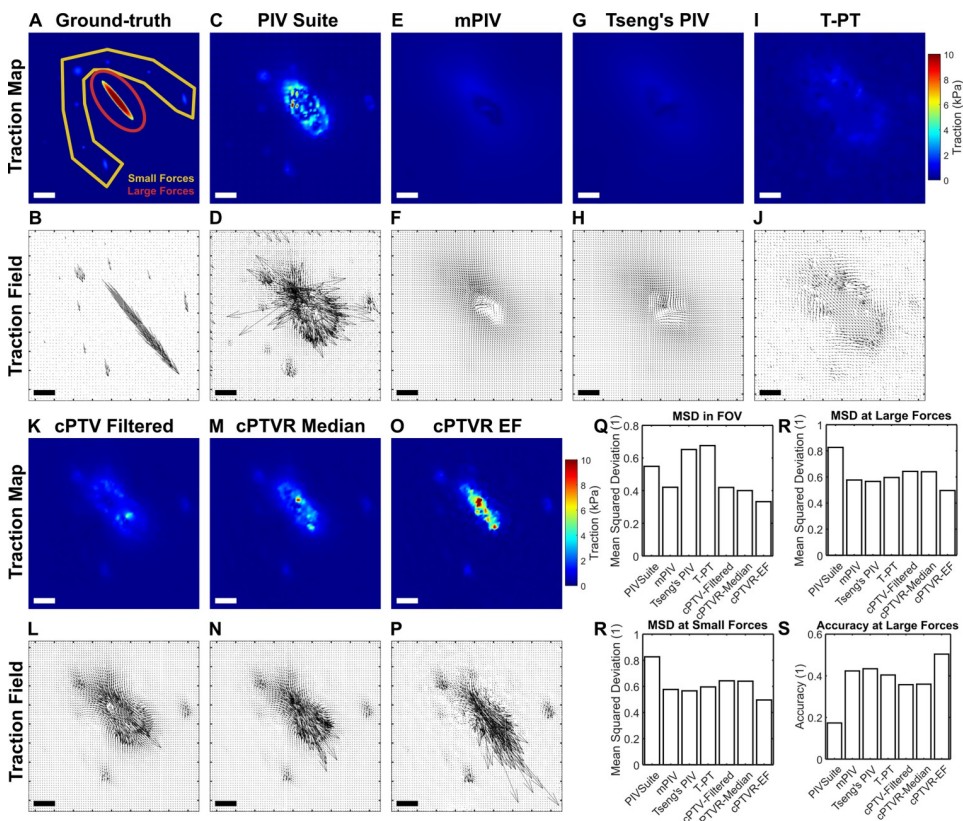

**Fig 5. cPTV-retracking method with the enlargement factor leads to the most accurate traction reconstruction for large forces.** (A) The ground-truth traction map, the same as in Fig 1A. (B) The ground-truth traction vector field. (C-P) the traction maps (C,E,G,I,K,M,O) and the traction vector fields (D,F,H,J,L,N,P) reconstructed from displacement fields by PIV Suite (C,D), mpiv (E,F), Tseng's PIV (G,H), T-PT (I,J), cPTV with outlier filtering (K,L), cPTV-Retracking with a median model vector (M,N), and cPTV-Retracking with the enlargement factor (O,P). (Q-S) Bar plots of the mean-squared-diviation (MSD) for the measured force fields by the six tracking methods quantified over the entire field of view (Q), large force area (R), and small force area (S). (T) A bar plot of accuracy for the measured force fields by the six tracking methods over the large force area.

vectors (Fig 5O and 5P). Quantifying the mean-squared deviation confirmed that cPTVR with the enlargement factor, or cPTVR-EF was able to produce the traction field with least deviation from the ground truth traction field (Fig 5Q). The lowest MSD exhibited by cPTVR-EF was attributed to the reduced deviation over the large force region (Fig 5R) as well as over the region of small forces (Fig 5S). Quantification of the vector accuracy also confirmed that cPTVR-EF-based traction produced the most accurate field compared to other methods (Fig 5T). Together, the traction reconstruction and comparison suggest that cPTVR-EF enables better detection of the large forces by being able to detect the large, local displacement vectors.

To test whether cPTVR-EF, which we refer to just cPTVR from now on, can track an experimental bead images for TFM, we conducted a TFM experiment with human bone osteosarcoma epithelial (U2OS) cells on a 4 kPa silicone gel coated with 40 nm-diameter fluorescent beads. We collected the bead images under total internal reflection fluorescence microscope when cells were adhered to the gel surface and after they are released (Fig 6A). The pair of the bead images were processed with PIVSuite, cPTV and cPTVR for displacement measurement (Fig 6B–6D). We picked PIVSuite as a representative PIV method based on the average performance shown compared to other PIV methods (Fig 1O–1Q). The displacement tracked by PIV Suite showed mostly smooth field except for the failure at one corner (Fig 6B). The

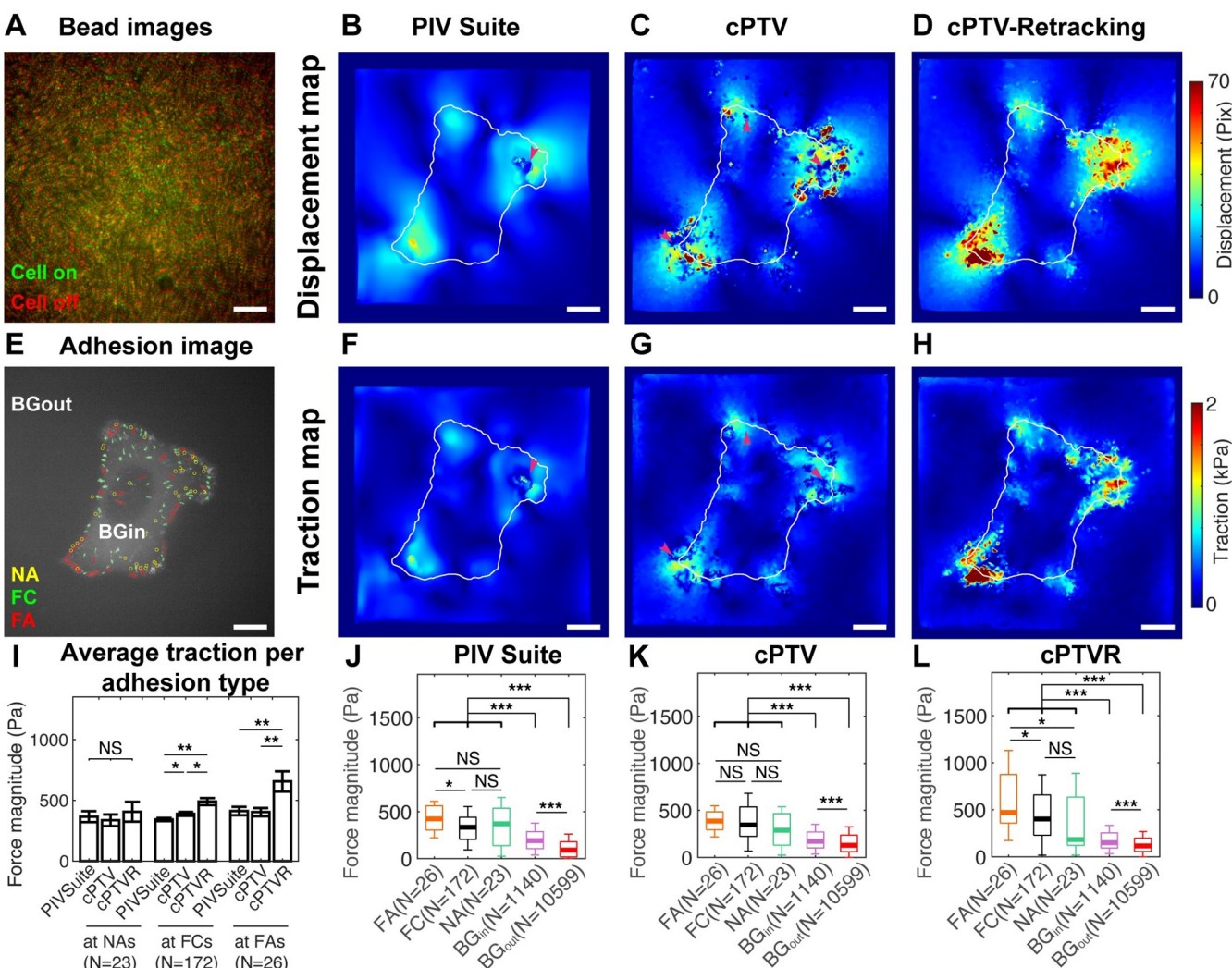

**Fig 6. cPTVR leads to a traction field resolving a larger range of the force magnitude than PIV or cPTV.** (A) A color-merged image of beads on a 4 kPa silicone gel, taken under cell presence (green) and absence (red) via total internal reflection fluorescence microscope. Scale bar: 5 μm (B-D) Displacement maps tracked by PIV Suite (B), cPTV (C) and cPTVR with the enlargement factor (D). (E) An image of paxillin-GFP of a U2OS cell, overlaid with nascent adhesions (NAs, yellow circles), detected with Gaussian misture model, focal complexes (FCs, green segmentations), and focal adhesions (FAs, red segmentations). Background inside a cell (BG$_{in}$) represents the area in the cell excluding adhesions whereas background outside the cell (BG$_{out}$) represents the area outside of the cell. (F-H) Traction mpas reconstructed from the displacement fields in B,C,D, respectively. White outlines overlaid on the maps represents cell boundary. Magenta arrowheads indicate areas of tracking failure. Scale bar: 5 μm (I) A bar graph of the average traction magnitude by PIV Suite, cPTV and cPTVR per NAs, FCs and FAs. Note that larger forces are meausred for FCs and FAs by cPTVR compared to the other two methods whereas forces at NAs stay the same. **: p<0.01, *: p<0.05, tested by unpaired student t-test. Scale bars: 5 μm. (J-L) Box plots of tractions quantified at FAs, FCs, NAs, BGin and BGout by PIV Suite (J), cPTV (K), and cPTVR (L). Note that only by cPTVR-based traction can resolve forces at FAs larger than forces at both FCs and NAs whereas other methods result in the force at FAs not necessarily larger than forces at either or both of FCs and NAs. ***: p<1e-5, *: p<0.5, tested by unpaired student t-test.

displacement by cPTV showed patches of large (>40 pixel in magnitude) displacements (Fig 6C). However, it showed even more places missing displacement tracking (Fig 6C, *magenta arrowheads*), consistent with the failure observed in the simulation study (Fig 1M and 1N). When cPTVR was used to quantify the displacement field, there were many more large displacement vectors detected, mostly near the cell edge (Fig 6D).

To assess the reconstructed traction fields' reasonability, we transfected the U2OS cells with paxillin-GFP, imaged its fluorescence under TIRF, and segmented and detected focal adhesions (FAs), focal complexes (FCs) and nascent adhesions (NAs) as done previously [2, 33].

The traction maps reconstructed from the displacement fields showed generally similar characteristics as in the displacement fields (Fig 6F–6H). Specifically, the smooth displacement field by PIV Suite resulted in the smooth, likely underestimated, traction field with missing traction vectors on one location (Fig 6F, *a magenta arrowhead*). The traction from the displacement tracked by cPTV showed a bit more detailed, and less underestimated field. However, the missing displacement vectors caused the tractions to be less organized by containing significant forces outside the cell boundary and positions of voids (Fig 6G, *magenta arrowheads*). Contrary to this, the traction from the displacement field by cPTVR contained much larger, e.g., >1.5 kPa, tractions with more coherent-looking traction distribution (Fig 6H). Comparison of the traction magnitude among the three traction fields revealed that whereas tractions at NAs were insignificantly different, the tractions at FCs and FAs were higher for cPTVR-based traction compared to both of the other two methods (Fig 6I), demonstrating cPTVR's outstanding performance in restoring large forces. Plotting the traction separately per tracking method revealed that whereas there is no or only a little difference in traction between FAs, FCs and NAs by PIV Suite or cPTV, there was a significant difference in traction at FAs compared to ones at FCs and NAs when cPTVR was used (Fig 6L). Altogether, these data suggest that cPTVR method enables resolving large traction forces transmitted by mature focal adhesions on a soft substrate by being able to rescue locally large displacement vectors.

## Discussion

In this study, we present a significantly improved deformation tracking method called correlation-based particle tracking velocimetry with retracking (cPTVR). We show that the existing PIV methods and the conventional cPTV are not able to track large, local deformation because the image correlation becomes insignificant at such displacements. Identifying the missing vectors are usually large vectors with orientation similar to their neighboring vectors, we implemented the new cPTVR method to use the neighbors to limit the interrogation range and to choose the correlation peak that is at least, or more than, the median of the neighboring displacement vectors. We show that cPTVR can appropriately infer locally large displacement vectors when used with the enlargement factor that gradually increases the magnitude of the model vector. The displacement field tracked by cPTVR results in the traction field that restores the large tractions. Using experimental bead images, we also show that cPTVR leads to overall larger displacement and traction by large focal adhesions of a cell on a relatively soft substrate.

One of the reasons behind conventional PIVs' failure for the large, local displacement might be related to the fact that the method has been developed to estimate fluid motions of various types of flows based on the images consecutively taken by a high-speed camera [12]. In that situation, the frame-to-frame displacement can be controlled to be trackable enough by using low-enough time-interval of the camera. However, such control is not allowed in TFM experiments as the image pair has to be taken before and after cell adhesion, which can contain a wide range of spatially-variable deformations. Our data show that with iterative process with neighbor finding and using them as a model vector can allow to pick a local correlation peak although that is not necessary a global maximum.

In the aspect of 'inferring' unknown displacement vectors, the proposed method is a bit similar to a model-based TFM method where the traction field is inferred via not only displacement field but images of F-actin and focal adhesions [34]. Our method is however different from this method as it relies only on the bead images to rescue initially missing displacement vectors using neighboring accurately-measured vectors. Our method is inspired by the 3D image interrogation method called fast-iterative digital volume correlation (FIDVC)

which iteratively progresses interrogation while performing image deformation [35]. A difference of our method from FIDVC is that 1) cPTVR intentionally avoids potential artifact caused by image warping; and 2) cPTVR starts its interrogation from the center points of individually identified beads whereas FIDVC, as other PIVs, uses points at the grid, which inherently can contain errors associated with off-centered beads in a template window [17].

As cPTVR tends to overestimate the missing displacements compared to neighbors, the retracking option is recommended to be used when the quality of the bead image has a sufficiently high signal-to-noise ratio, and when bead movements are within the same focal plane, i.e., minimal out-of-plane motions. This way the algorithm would be able to function properly based on well-tracked initial displacement vectors.

Altogether, we present a method that can help improve the TFM accuracy especially situation where locally large deformation is encoded in the bead images due to large forces. Our retracking method is not limited to the correlation-based PTV, but is easily adoptable to other recent methods such as feature-vector based displacement tracking [27, 36] or optical tracking method [5].

## Software

The TFM package which includes cPTVR is available via Han Lab website at hanlab.biomed.mtu.edu/software, or via GitHub at github.com/HanLab-BME-MTU/TFMPackage.git.

## Supporting information

**S1 Fig. Sensitivity test of key parameters in PIV suite.** In this test, the template size, a kind of filter, and the grid spacing of PIV Suite, as a representative of PIV methods, was tested on the bead images containing intermediate displacement. (A) Ground-truth traction map showing intermediate level (~100 Pa) of traction in the center. All traction vectors (invisible) are directing downward, i.e., negative y-direction. (B) Ground-truth displacement magnitude map calculated using Boussinesq solution with E = 1kPa. (C) A color-merged image of a synthetic bead image (8000 beads on 512x512 pixel area) in a relaxed configuration (red) and another image of beads where displacements in A is applied (green). Scale bars in A-C: 1 μm assuming 108 nm/pixel. (D-F) Bar plots of the mean-squared-deviation (MSD) for the measured displacement fields by PIV Suite with different template window lengths (D), template filters (E) and grid spacing or window step size (F). Parameters not being tested were maintained as in Table 1. For panel F's x-label, 1X means that the step size is the same length as template window length, i.e., 16 pixel in the case. These tests demonstrate that small-enough template window, step size and Welch filter produce the best tracking results.
(TIF)

## Acknowledgments

We acknowledge Jiri Vejrazka (Institute of Chemical Process Fundamentals ASCR, Prague, Czech Republic), an author of PIV Suite, Nobuhito Mori (Kyoto University, Japan) and Kuang-An Chang (Texas A&M University), authors of mpiv, Qingzong Tseng (EMBL, Germany), an author of Tseng's PIV and Mohak Patel and Christian Franck (UW Madison), authors of T-PT for their software to be compared with our cPTVR method. We also thank Alex Groisman (UCSD) for providing soft substrates for experimental TFM experiments.

## Author Contributions

**Conceptualization:** Sangyoon J. Han.

**Data curation:** Samuel E. Haarman, Sue Y. Kim, Sangyoon J. Han.

**Formal analysis:** Samuel E. Haarman, Sue Y. Kim.

**Funding acquisition:** Sangyoon J. Han.

**Methodology:** Samuel E. Haarman, Sangyoon J. Han.

**Resources:** Tadamoto Isogai, Kevin M. Dean.

**Software:** Sue Y. Kim.

**Visualization:** Sue Y. Kim.

**Writing – original draft:** Sue Y. Kim.

**Writing – review & editing:** Sangyoon J. Han.

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
