## [Decision Letter · Decision Letter 0]

2 Feb 2022

PONE-D-21-28667Particle Retracking Algorithm Capable of Quantifying Large, Local Matrix Deformation for Traction Force MicroscopyPLOS ONE

Dear Dr. Han,

Thank you for submitting your manuscript to PLOS ONE. After careful consideration, we feel that it has merit but does not fully meet PLOS ONE’s publication criteria as it currently stands. Therefore, we invite you to submit a revised version of the manuscript that addresses the points raised during the review process.

Please address the reviewer comments. In addition, please response and address the following comments:

Page 10, line 3: please revise the term ‘high-speed’; PIV utilizes either high or slow speeds cameras, depending on the application.

Page 10, line 8: the pixel to micro meter ratio is determined by the focal length of the microscopes and lenses, this is a parameter that can be changed using different apparatus. Thus; 40 pixel is a private case

The authors based their suggested algorithm on existing particle tracking one and perform a comparison with current open-source PIV platforms.

Many parameters are missing from the comparison: was the interrogation window a square one with a length of 21 pixels? Was multipass applied, what were the filters values, where there any validation and interpolation performed on the correlation, is the correlation engine based on direct Fourier transformation or was it direct correlation? All these parameters may impact the results obtained

Second, 21-pixel grid size, if this corresponds to the interrogation window length, then assuming DFT was performed, all the correlations were based on 32X 32 (2^5^) with zero padding to complete the matrix. In essence, it is unclear the reasoning for choosing this interrogation window. If the goal was to capture only 2 beads and not more, then PIV technique by its definition will not generate meaningful results because as the authors mentioned, it is an average-based concept where the motion of small clustered particles is obtained, assuming that the side of the window is comparable to the small scale of the phenomena.

Lastly, the authors choose to compare with LabPIV, mPIV and Tsueng’s PIV; all of these available open-source platforms were not peer-reviewed in accredited journals. Some of these platforms consist of biased errors that not necessarily generates reliable output. It is highly recommended to perform a parametric analysis using different interrogation window and filtering such that the concept presented herein will be validated properly.

We look forward to receiving your revised manuscript.

Kind regards,

Roi Gurka

Academic Editor

PLOS ONE

Journal Requirements:

This work was funded by NIH R15GM135806 (S.J.H.) and NIH F32GM117793 (K.M.D.).

This work was funded by National Institutes of Health (https://www.nih.org/) R15GM135806 (S.J.H.) and F32GM117793 (K.M.D.). The funders had no role in study design, data collection and analysis, decision to publish, or preparation of the manuscript.

Reviewers' comments:

Reviewer's Responses to Questions

**Comments to the Author**

1. Is the manuscript technically sound, and do the data support the conclusions?

Reviewer #1: Yes

2. Has the statistical analysis been performed appropriately and rigorously? 

Reviewer #1: Yes

3. Have the authors made all data underlying the findings in their manuscript fully available?

Reviewer #1: Yes

4. Is the manuscript presented in an intelligible fashion and written in standard English?

Reviewer #1: Yes

5. Review Comments to the Author

Reviewer #1: Cell-generated matrix deformations are typically detected using particle image velocimetry (PIV) or correlation-based particle tracking velocimetry (cPTV). Here, Kim et al. developed a cPTV-Retracking algorithm that enables a more accurate and robust detection of large matrix-deformations.

This cPTV-Retracking algorithm performs a significance filtering to detect poorly matched, large matrix deformations, and then re-tracks these regions by using information from their well-tracked neighbours. The missing matrix deformations are assumed to have similar directionality and amplitude as their neighbouring deformations. Local maxima of the correlation score are computed in a reduced FoV with regards to the model vector (average of neighbouring vectors). If the significance criterion is still not fulfilled, the local maxima are compared to a second model vector (median of neighbouring vectors). If still no match was found, the template window size is iteratively lowered (and optionally the amplitude of the model vector is increased) until a deformation vector in proximity to the position passes the significance criterion.

Based on simulated traction fields (~10 kPa), the cPTV-Retracking algorithm achieves a more accurate detection for large deformations compared to classical particle image velocimetry (PIV) or correlation-based particle tracking velocimetry (cPTV).

Major

To compare different PIV/cPTV-methods, the parameters such as window size and overlap must be chosen similarly. The method section states that the same grid-size (2.2µm or 21 pixels) was used for all methods, however, the different resolution/grid-spacing of the displacement & traction fields shown in Fig.1&5 suggest that different settings might be used for the different PIV/cPTV methods. If this is the case, the authors need to compare the methods using similar parameters, or else they need to render the figures with the same density of deformation vectors.

Additionally, a window-size of 21px might not be suited for maximal deformations of 60px. For a comparison with other methods, it would be important that the authors also test smoother deformation fields (that more closely resemble the typical deformation fields around cells), or increase the window sizes and overlap.

The model vector from neighbouring vectors is initially calculated using the mean value and - if no matching deformation is found - the median value from the neighbouring vectors. The authors state the median value is better suited since the mean value tends to overestimate deformations. Are there reasons why the median model vector is not used right from the beginning?

In PIV-methods, outliers are commonly filtered by their Signal-to-Noise ratio and replaced by the average of neighbouring vectors. This approach is similar to cPTVr, and it would be important to see how cPTVr compares to PIV+filtering.

Minor

“Bead images were entered into TFM package to be processed for displacement tracking and traction reconstruction.” Which package was used?

The following statements appear to be overstated: “all current PIV and cPTV methods fail to track a large, local displacement field” or “..large displacement caused by a large local traction is unable to be tracked via cross-correlation..”

Top part of the text in Fig. 3b is cropped and not readable.

The numbering in the figure legends below the main-text does not match the numbering of the figure legends in the main-text.

The used “significance criterion to produce only well-tracked vectors” should be described, at least briefly

6. PLOS authors have the option to publish the peer review history of their article (what does this mean?). If published, this will include your full peer review and any attached files.

Reviewer #1: No

---

## [Author Response · Author response to Decision Letter 0]

21 Apr 2022

Responses to Editor

1. Page 10, line 3: please revise the term ‘high-speed’; PIV utilizes either high or slow speeds cameras, depending on the application.

We appreciate this comment. We agree that PIV does not necessarily work only with high-speed camera. We removed the term from the manuscript.

2. Page 10, line 8: the pixel-to-micrometer ratio is determined by the focal length of the microscopes and lenses, this is a parameter that can be changed using different apparatus. Thus; 40 pixel is a private case.

We agree with the editor’s point. 40 pixel is only possible when we assume a 60x objective with a camera with 6.5 micrometer/pixel of the pixel size, which results in usually 0.108 micrometer/pixel of the final resolution. To clear the subjectiveness, we removed the ’40 pixel’ from the revised manuscript. 

3. The authors based their suggested algorithm on existing particle tracking one and perform a comparison with current open-source PIV platforms. Many parameters are missing from the comparison: was the interrogation window a square one with a length of 21 pixels? Was multipass applied, what were the filters values, where there any validation and interpolation performed on the correlation, is the correlation engine based on direct Fourier transformation or was it direct correlation? All these parameters may impact the results obtained

We are grateful for the editor’s concern. We realize that we have not fully disclosed information about interrogation parameters used for the three PIV platforms. Now we clearly mentioned such parameters and relevant options in the method section of the manuscript. Briefly, all PIV methods used iterative, multi-pass tracking with 64, 64, 32, 16 pixels in template length. A Welch filter was used because it provides improvement compared to other filters. The cross-correlation was performed in the frequency domain using Fourier transform. We tested these parameters using bead images with small deformation. 

4. Second, 21-pixel grid size, if this corresponds to the interrogation window length, then assuming DFT was performed, all the correlations were based on 32X 32 (25) with zero padding to complete the matrix. In essence, it is unclear the reasoning for choosing this interrogation window. If the goal was to capture only 2 beads and not more, then PIV technique by its definition will not generate meaningful results because as the authors mentioned, it is an average-based concept where the motion of small, clustered particles is obtained, assuming that the side of the window is comparable to the small scale of the phenomena.

We appreciate this comment. We realize that our description about the interrogation window size was confusing. The 17 pixel was used for the interrogation window length only in cPTV, whereas 32x32 or 16x16 were used for PIV methods. Our philosophy to go to smaller window length, e.g., 16x16 pixel, was to achieve better accuracy by avoiding the averaging effect from a larger interrogation area and to be consistent with parameters used for cPTV. Since we used an multipass interrogation with progressive, iterative deformation of the image, we believe that a global, low-frequency deformation is already captured during the early-pass interrogation with larger interrogation window, after which a more detailed, high-frequency, deformation is measured by a smaller interrogation window at the next iteration in PIV methods. Our sensitivity test results also show that 16 pixel window length in final iteration produces the most accurate tracking outputs. Now we described the parameters for PIVs and cPTV more clearly, i.e., by providing the window length and the justification about why at the methods section.

5. Lastly, the authors choose to compare with LabPIV, mPIV and Tseng’s PIV; all of these available open-source platforms were not peer-reviewed in accredited journals. Some of these platforms consist of biased errors that not necessarily generates reliable output. It is highly recommended to perform a parametric analysis using different interrogation window and filtering such that the concept presented herein will be validated properly.

We agree with the editor’s concern. The reason why we used the open-source platforms is the accessibility of the software. We identify there are still many PIV or optical flow algorithms (or combined approaches) that are published, e.g., Theunissen et al., 2009, Exp in Fluids, Schmidt and Sutton 2020 Exp in Fluids, and Seong et al., 2019, Exp in Fluids, etc. Most of the times, however, the codes were not shared in those published articles. To get rid of potential concerns about biases in the open-source algorithms, we conducted a sensitivity study with a range of interrogation window lengths, grid spacing and filters for an image pair reflecting a modest level of deformation. We showed the results in S1 Fig and listed the best parameters in Table 1 of the manuscript. Furthermore, to involve a published method, we now included results from topology-based particle tracking (T-PT) (Patel et al., 2018, Sci Rep). Although it is not under a category of PIV, T-PT is believed to be an efficient tracking method for gel deformation used for TFM purpose. Nonetheless, our test shows that T-PT also fails to track the very large, local deformation. We updated the manuscript accordingly with the test of this new result.

Responses to Reviewer 1

Major

1. To compare different PIV/cPTV-methods, the parameters such as window size and overlap must be chosen similarly. The method section states that the same grid-size (2.2 µm or 21 pixels) was used for all methods, however, the different resolution/grid-spacing of the displacement & traction fields shown in Fig.1&5 suggest that different settings might be used for the different PIV/cPTV methods. If this is the case, the authors need to compare the methods using similar parameters, or else they need to render the figures with the same density of deformation vectors.

We appreciate the reviewer’s note. We agree that we might have used different grid spacing in some PIV methods. We have now gone through all the parameters used per PIV and cPTV, including grid spacing, and have made sure that they are within similar range across the tracking methods. We updated the method section with new Table 1, S1 Fig, Figs 1 and 5 of the manuscript in our revised version. 

2. Additionally, a window-size of 21px might not be suited for maximal deformations of 60px. For a comparison with other methods, it would be important that the authors also test smoother deformation fields (that more closely resemble the typical deformation fields around cells), or increase the window sizes and overlap.

We appreciate this comment. We realize that our description about the interrogation window size was confusing. The 17-21 pixel was used for the interrogation window length only in cPTV, whereas 32x32 or 16x16 were used for PIV methods. As the editor mentioned, PIV technique by its definition cannot generate a meaningful result. Our philosophy to go to smaller window length, e.g., 16x16 pixel, was to be consistent with parameters used for cPTV and to achieve better accuracy by avoiding the averaging effect from a larger interrogation area. Since we used an multipass interrogation with progressive deformation of the image, we believe that a global, low-frequency deformation is already captured during the early-pass interrogation, from which a more detailed, high-frequency, deformation can be measured by a smaller interrogation window at the next iteration in PIV methods. Now we described the parameters for PIVs and cPTV more clearly, i.e., by providing the window length and the justification about why at the methods section. Additionally, to make sure we are using the best parameters for testing, we conducted a sensitivity study with a range of interrogation window lengths for an image pair reflecting a modest level of deformation. The results are now included as a supplementary information, which has shown that a larger window length (in the final pass) leads to systematic underestimation of the deformation and that the current window length (16 pixel) guarantees the best accuracy. 

3. The model vector from neighbouring vectors is initially calculated using the mean value and - if no matching deformation is found - the median value from the neighbouring vectors. The authors state the median value is better suited since the mean value tends to overestimate deformations. Are there reasons why the median model vector is not used right from the beginning?

We are thankful for the reviewer’s comment. We think that our description about the model vector was confusing. We have not used the mean vector as a model vector at all, but we accidentally described it in the legend of Fig. 3. The algorithm uses the median model vector right from the beginning if the correlation-score-based selection fails. The reason why we compare with the model vector only after the correlation score-based selection is failed is that we give an opportunity for a vector with the high correlation score first. We have corrected the description in the Fig. 3 legend and the description. 

4. In PIV-methods, outliers are commonly filtered by their Signal-to-Noise ratio and replaced by the average of neighbouring vectors. This approach is similar to cPTVr, and it would be important to see how cPTVr compares to PIV+filtering.

We totally agree with the reviewer’s comment. PIV’s outliner filtering and interpolation was the main motivation for cPTVR. We used the option of PIV’s outlier filtering + filling with neighbor-based interpolated vectors in our usage of them in Fig. 1, which showed failure of tracking for the large displacement vectors. 

Minor

1. “Bead images were entered into TFM package to be processed for displacement tracking and traction reconstruction.” Which package was used?

Thank you for catching the ambiguous expression. The TFM package used was the software we have published before (Han et al., 2015 Nature Methods, Mittal and Han, 2021, Cur Protocols). We have now cited these papers for the package. 

2. The following statements appear to be overstated: “all current PIV and cPTV methods fail to track a large, local displacement field” or “..large displacement caused by a large local traction is unable to be tracked via cross-correlation..”

Thank you for pointing these out. Those statements are according to our simulation results from Fig. 1. Not all large displacements are meant to be unable to be tracked, but the ones with large spatial gradient are shown to be not trackable due to poor correlation. We have now toned down for those statements.

3. Top part of the text in Fig. 3b is cropped and not readable.

We unfortunately were not able to find the same cropped image in our submitted manuscript. However, to avoid potential problem, we are submitting individual figure image files, including Fig. 3., and doublechecked that they are not cropped any sides.

4. The numbering in the figure legends below the main-text does not match the numbering of the figure legends in the main-text.

We went over figure citations in the main text and made sure that they match with the panel labeling in the figure and its legends. We appreciate the reviewer’s careful reading.

5. The used “significance criterion to produce only well-tracked vectors” should be described, at least briefly.

We are again thankful for the reviewer’s note. We completely agree with the reviewer that the significance criterion needs to be defined. We added the definition in the methods section.

---

## [Editor Report · Decision Letter 1]

4 May 2022

Particle Retracking Algorithm Capable of Quantifying Large, Local Matrix Deformation for Traction Force Microscopy

PONE-D-21-28667R1

Dear Dr. Han,

We’re pleased to inform you that your manuscript has been judged scientifically suitable for publication and will be formally accepted for publication once it meets all outstanding technical requirements.

Kind regards,

Roi Gurka

Academic Editor

PLOS ONE
---

## [Editor Report · Acceptance letter]

26 May 2022

PONE-D-21-28667R1 

Particle Retracking Algorithm Capable of Quantifying Large, Local Matrix Deformation for Traction Force Microscopy 

Dear Dr. Han:

I'm pleased to inform you that your manuscript has been deemed suitable for publication in PLOS ONE. Congratulations! Your manuscript is now with our production department. 

Kind regards, 

on behalf of

Dr. Roi Gurka 

Academic Editor

PLOS ONE